# Effective electrical manipulation of a topological antiferromagnet by orbital torques

Zhenyi Zheng[1,6], Tao Zeng[1,6], Tieyang Zhao [1,6], Shu Shi[1], Lizhu Ren[2], Tongtong Zhang[3], Lanxin Jia[1], Youdi Gu[1], Rui Xiao[1], Hengan Zhou[1], Qihan Zhang[1], Jiaqi Lu[1], Guilei Wang[4], Chao Zhao[4], Huihui Li[4]✉, Beng Kang Tay[3]✉ & Jingsheng Chen[1,5]✉

The electrical control of the non-trivial topology in Weyl antiferromagnets is of great interest for the development of next-generation spintronic devices. Recent studies suggest that the spin Hall effect can switch the topological antiferromagnetic order. However, the switching efficiency remains relatively low. Here, we demonstrate the effective manipulation of antiferromagnetic order in the Weyl semimetal $Mn_3Sn$ using orbital torques originating from either metal Mn or oxide $CuO_x$. Although $Mn_3Sn$ can convert orbital current to spin current on its own, we find that inserting a heavy metal layer, such as Pt, of appropriate thickness can effectively reduce the critical switching current density by one order of magnitude. In addition, we show that the memristor-like switching behaviour of $Mn_3Sn$ can mimic the potentiation and depression processes of a synapse with high linearity—which may be beneficial for constructing accurate artificial neural networks. Our work paves a way for manipulating the topological antiferromagnetic order and may inspire more high-performance antiferromagnetic functional devices.

Topological materials have attracted intensive attentions due to their robust topologically protected states, many exotic properties and promising applications for quantum computing and spintronics[1,2]. According to the dimensionality of electronic bands touching, the topological states of materials can be classified into topological insulators[3,4], Dirac semimetals[5,6] and Weyl semimetals[7,8], etc. Weyl semimetal has the feature of Weyl fermion with the presence of the chiral node (i.e. Weyl node) and the Fermi arc surface states connecting the Weyl-node pair with opposite chirality[9]. The Weyl node is a linearly crossing point of two non-degenerate bands which requires breaking inversion symmetry or time reversal symmetry. In order for developing electronic device, it is essential for effective electrical

manipulation of the nontrivial topologic states e.g. Weyl nodes. Magnetic Weyl semimetal is considered as an ideal material candidate since the time reversal symmetry is breaking and the location and enegy of Weyl nodes in the Brillouin zone depend on the magnetization direction[2]. $Mn_3Sn$ is a typical Weyl semimetal and non-collinear antiferromagnet (AFM)[10–13]. As shown in Fig. 1a, the spin structure of $Mn_3Sn$ consists of two kagome planes with opposite chirality. This hexagonal spin texture can be considered as a ferroic ordering of a cluster magnetic octupole $M$ and it breaks time reversal symmetry macroscopically. AFMs have negligible stray field and ultra-fast magnetic dynamics, which helps to overcome the integrability and speed bottlenecks of traditional spintronic devices[14–17]. Furthermore, all-AFM-

[1]Department of Materials Science and Engineering, National University of Singapore, Singapore 117575, Singapore. [2]Department of Electrical and Computer Engineering, National University of Singapore, Singapore 117575, Singapore. [3]Centre for Micro- and Nano-Electronics (CMNE), School of Electrical and Electronic Engineering, Nanyang Technological University, 639798 Singapore, Singapore. [4]Beijing Superstring Academy of Memory Technology, Beijing 100176, China. [5]Chongqing Research Institute, National University of Singapore, Chongqing 401120, China. [6]These authors contributed equally: Zhenyi Zheng, Tao Zeng, Tieyang Zhao. ✉e-mail: lihh04@163.com; ebktay@ntu.edu.sg; msecj@nus.edu.sg

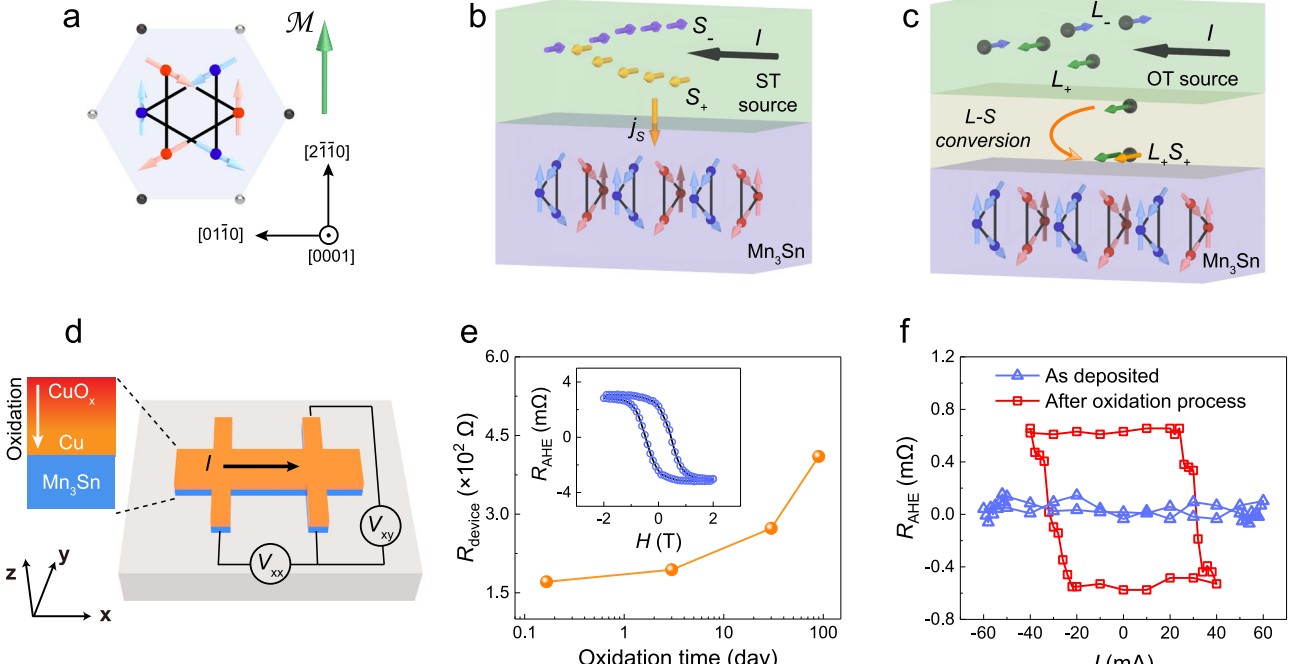

**Fig. 1 | Schematic of OT-driven magnetization switching in Mn₃Sn. a** Spin structure of Mn₃Sn. The large blue and red circles (small black and silver circles) represent Mn (Sn) atoms. In two kagome planes with different chirality, blue and red arrows indicate the magnetic moments of Mn atoms in different layers. The green arrow indicates the $M$ direction of the formed cluster magnetic octupole. **b** Schematic of ST-driven magnetization switching in Mn₃Sn. Current in spin source layer can generate spin angular momentum $S_{+(-)}$ which is injected into Mn₃Sn layer and exerts a torque on the kagome planes of Mn₃Sn. **c** Schematic of OT-driven magnetization switching. Before being exerted on Mn₃Sn, orbital angular momentum $L_{+(-)}$ induced by current in OT source layer needs to be converted to $S_{+(-)}$. **d** Stack structure of Mn₃Sn/Cu/CuOₓ film and electrical measurement setup. **e** Longitudinal resistance of Mn₃Sn/Cu device as a function of oxidation time. Inset illustrates $R_{AHE}$ versus applied magnetic field. **f** Current-induced magnetization switching loops in as-deposited Mn₃Sn/Cu device and in Mn₃Sn/Cu/CuOₓ device.

based magnetic tunnel junctions (MTJ) with a sizable tunneling magnetoresistance (TMR) ratio have recently been demonstrated[18,19].

To date, researchers have demonstrated that spin torques (ST) can manipulate the Weyl nodes in Mn₃Sn or Co₂MnGa manifested with the change in anormolous Hall effect (AHE)[20–24], using a similar protocol as for heavy metal/ferromagnets (HM/FM)[25–29] where the spin current generated in heavy metal by spin Hall effect (SHE) is injected into FM and a torque is exerted on FM. As shown in Fig. 1b, the generated spin current induced by current along <0001> direction can be directly exerted on the kagome planes of Mn₃Sn and induce the magnetization switching. The switching efficiency largely depends on the charge-current-to-spin-current conversion efficiency, i.e., the spin Hall angle (SHA), in the adjacent spin current source layer. To improve the efficiency and reduce the switching current density $J_c$, except for exploring novel materials with high SHA, it is also desirable to search alternative mechanisms to realize current-induced switching of topological states with high energy efficiency.

In this work, we propose to utilize orbital torques (OT) which originate from orbital Hall effect (OHE) or orbital Rashba-Edelstein effect (OREE) to manipulate the magnetic order in topological AFM. The basic schematic is shown in Fig. 1c. An applied current along <0001> direction can induce orbital current in OT source layer[30–32]. Before the generated orbital current can be exerted on the magnetization of Mn₃Sn, it need to be converted to spin current by an additional spin-orbit coupling (SOC)[33]. Compared with the limited SHA in spin current source material, it has been demonstrated that orbit current source materials possess a much higher orbit current generation efficiency[34,35]. Therefore, the critical switching current density $J_c$ is expected to be effectively reduced. Herein, we successfully demonstrate OT-driven magnetization switching in Mn₃Sn and prove that the orbit-current-to-spin-current (L-S) conversion can be done either by Mn₃Sn itself or by inserting a heavy metal with strong SOC.

we have achieved $J_c$ as low as -1 × 10¹⁰ A/m², which is more than one order of magnitude lower than the common $J_c$ in SHE-driven framework. Furthermore, we show that the stable memristor-like switching characteristics offers Mn₃Sn with excellent plasticity to mimic an artificial synapse with linear potentiation and depression processes, which is beneficial for constructing neural networks with high accuracy.

## Results

### OT-driven magnetization switching in topological Mn₃Sn

We deposited 40-nm-thick Mn₃Sn film on thermally oxidized silicon substrates by magnetron sputtering (see details in Methods). By energy dispersive spectrometry (EDS) mapping, the atomic percentage of MnSn alloy is determined to be around Mn₍₃.₀₅₋₃.₁₎Sn. The x-ray diffraction (XRD) $\theta$–$2\theta$ scans results of the deposited Mn₃Sn is shown Supplementary Note 1. Compared with pure Si substrate, a clear Mn₃Sn (0002) crystal peak is observed in the film sample. SQUID measurement indicates that our film exhibits a tiny out-of-plane magnetization (see Supplementary Note 1). This tiny out-of-plane hysteresis loop suggests there exist crystalline grains with its kagome plane in the film normal since the spin canting is in the (0001) kagome plane. We further carried out magneto-transport and anormolous Nerst effect (ANE) measurements to confirm the Weyl semimetal of our deposited Mn₃Sn films. Planar Hall effect (PHE) and in-plane angular magnetoresistance (AMR) are shown in Supplementary Figure 1c. The PHE and AMR follow the functions $R_{xy} = -\Delta R \sin\theta \cos\theta$ and $R_{xx} = R_{\perp} - \Delta R \cos^2\theta$, respectively, where $\Delta R = R_{\perp} - R_{\parallel}$, and $R_{\perp}$ and $R_{\parallel}$ are the resistances when the magnetic field directions are perpendicular and parallel to the charge current direction, respectively. These are consistent with the feature of the chiral anomaly induced PHE and AMR in Weyl semimetal[9,20]. Comparing to the effect of magnetization, ANE in Weyl semimetal is much enhanced due to the Weyl nodes around Fermi level[11]. The large

ANE and small magnetization further confirm that our Mn₃Sn films are the Weyl semimetal (Supplementary Figure 1b and c).

It has been widely demonstrated that OT can be observed in the naturally oxidized Cu[36,37]. Thus, we firsly choose oxidized Cu to verify the impact of OT on the magnetization switching of Mn₃Sn antiferromagnet. As shown in inset of Fig. 1d, we firstly deposited a Cu layer on the Mn₃Sn layer and then follow the method in previous works to naturally oxidize Cu at atmosphere for different time[37]. The films were fabricated into Hall bar device of 5 μm width to implement magnetotransport measurement. The schematic setup of the measurement is illustrated in Fig. 1d. The longitudinal resistance shows a continuous increase with oxidation time, which suggests the gradual oxidation of Cu layer with the time. We then measure the AHE resistance $R_{AHE}$ (inset in Fig. 1e) as a function of the out-of-plane magnetic field to estimate the switchable magnetic domains which corresponds to the crystalline grains with Kogame plane in the film normal. In the absence of magnetic field, there exist two stable magnetic states which correspond to the magnetic octupole $M$ along ± z directions, respectively.

We then carry out current-induced switching experiments (Fig. 1f). Since Cu is a light metal with negligible SOC, current-induced magnetization switching is absent in the as-deposited Mn₃Sn/Cu film. As a comparison, a deterministic magnetization switching loop, corresponding to a switching ratio of ~ 25%, is well achieved in the device after natural oxidatization process of the Cu. We are also aware that several works reported current-induced switching in Mn₃Sn single layer with specific crystal configuration[38,39]. However, we didn't observe any switching phenomenon in our deposited Mn₃Sn single layer (Supplementary Note 2), verifying that the magnetization switching driving force indeed comes from the achieved Cu/CuOₓ layer. Additionally, in Supplementary Note 3, we demonstrated that only when the applied FM like Ni exhibits a high SOC, one can observe a sizable effective SHA in FM/Cu/CuOₓ heterostructure[40]. From one side, it verifies that the main switching driving force from Cu/CuOₓ is OTs instead of possible STs. From the other side, it emphasizes again that

the orbital current originated from CuOₓ must complete the $L$-$S$ conversion process shown in Fig. 1c to manipulate the magnetic dynamics in FM. Our deterministic switching results in Mn₃Sn/Cu/CuOₓ device directly prove that Mn₃Sn itself can complete the $L$-$S$ conversion process like what Ni does and the spin current converted from orbit current is then to switch the magnetization of Mn₃Sn layer. More experimental evidences related to SOC and $L$-$S$ conversion in Mn₃Sn will be presented in the next section.

Note that, for practical use, it is important to quantitatively control the orbital Hall angle in the device. In such case, metallic OT sources have an application edge over naturally oxidized Cu. We thus employ an heterostructure composed of Mn₃Sn (40 nm)/Pt($t_{Pt}$)/ Mn($t_{Mn}$) trilayer (Fig. 2a) to further investigate the detailed OT-based manipulation characteristics of topological magnetization. Mn is theoretically predicted to possess a large orbital Hall angle (~18)[35]. The inserted Pt layer serve as an additional $L$-$S$ conversion layer which helps to gain more spin torques[33,41]. In a device where $t_{Pt}$ = 2 nm and $t_{Mn}$ = 10 nm, deterministic OT-driven magnetization switching in Mn₃Sn is achieved as well (see detailed switching loops in Supplementary Note 4). An external magnetic field $H_{ex}$ is required to break the in-plane symmetry during the switching process. When reversing the direction of $H_{ex}$, the switching polarity also changes. This switching characteristic is very similar to the SHE-induced switching protocol for FMs. Moreover, we have observed a $H_{ex}$-dependent switching ratio trend in the sample. The switching ratio is defined as $\Delta R_c/\Delta R_H$, where $\Delta R_c$ and $\Delta R_H$ are current-induced and field-induced change of $R_{AHE}$. As shown in Fig. 2b, as the absolute value of $H_{ex}$ increases, the switching ratio will first increase and then saturate when exceeds 2 kOe. The saturated switching ratio is around 27%, which is comparable with other reported value[20,42,43]. This relatively small switching ratio can be explained by the fact that in-plane torques only allow the kagome planes to rotate between two energy minimum states with $\theta = ±\pi/6$ (see Fig. 2c) according to the symmetry analysis[20]. Note that all the following switching experiments in this work were carries out under

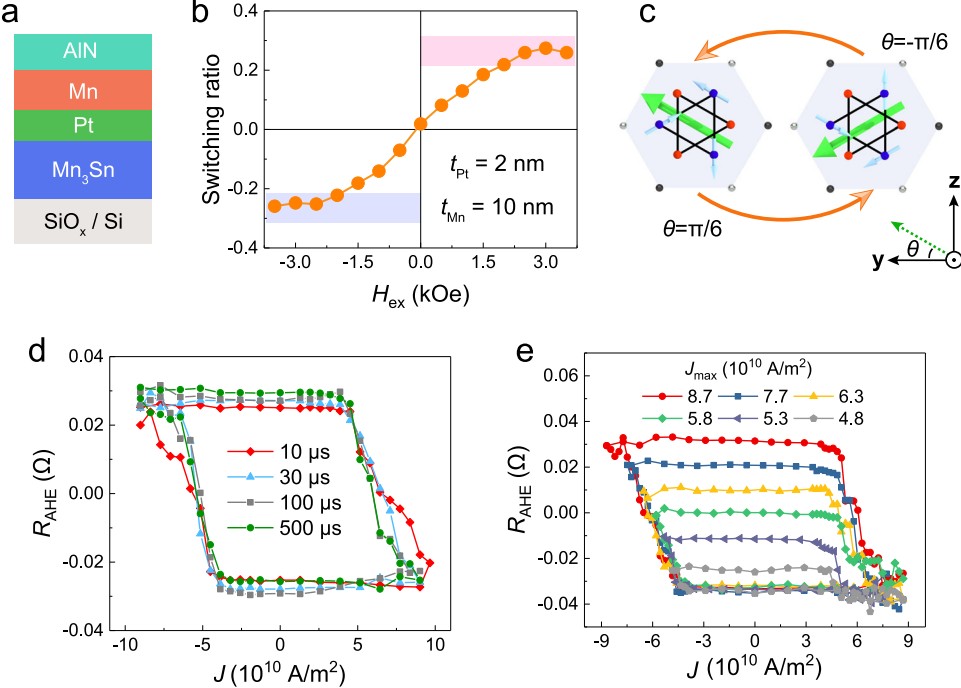

**Fig. 2 | OT-driven magnetization switching in Mn₃Sn/Pt(2 nm)/Mn(10 nm).**
**a** Schematic of the Mn₃Sn/Pt/Mn heterostructure. **b** OT-driven switching ratio as a function of $H_{ex}$. The ratio saturates when $H_{ex}$ exceeds 2 kOe. Red (purple) region indicates the saturated switching ratio area in the positive (negative) magnetic field

range. **c** Schematic of two stable magnetic states in Mn₃Sn during OT-driven switching dynamics. **d** $R_{AHE}$-$J$ switching loops with different applied pulse width. **e** Minor switching loops of the sample by limiting the maximum current value $J_{max}$ in negative range, as indicated by the numbers in the figure.

$H_{ex} = 2$ kOe, unless specified. Additionally, in Supplementary Note 5, we show that ANE signal switches simultaneously with the anomalous Hall signal, suggesting that non-trivial topology in $Mn_3Sn$ can also be manipulated by the orbital torque.

An important difference between switching in FM and switching in $Mn_3Sn$ is the impact of applied current pulse width. In FM system, a thermally activated switching model, in which $J_c$ decreases exponentially with the increasing pulse width, is widely accepted[44]. In Fig. 2d, we plot $R_{AHE}$ as a function of the applied charge current density $J$ in Pt/Mn bilayer with different pulse width. Clearly, $J_c$ in $Mn_3Sn$ is almost insensitive to the current pulse width which varies from 10 μs to 500 μs (see Fig. 2d). This insensitivity indicates that our deposited $Mn_3Sn$ possesses a good thermal stability, which is beneficial for the device scalability. We also notice that the achieved $R_{AHE}$-$J$ switching loops are quite 'tilted', i.e., there exists a series of intermediate states. As shown in Fig. 2e, a series of minor switching loops can be achieved by limiting the maximum value at negative current range. In Supplementary Note 6, we show that controlling the applied out-of-plane magnetic field can also achieve similar minor loops. The stable existence of these minor loops reveals that the intermediate states are non-volatile and can be recovered to the same initial state by applying a current density $J \sim 9 \times 10^{10}$ A/$m^2$. This switching characteristic indicates that the $Mn_3Sn$ device can possibly memorize the past electrical current pulse and be adapted as a memristor. We will further investigate the potential application of this memristor-like behavior in neuromorphic computing in the last section.

## OT source layer dependence of switching efficiency

To further verify that the observed switching behaviors are dominant by OTs in $Mn_3Sn/Pt/Mn$ and to optimize the switching performance, we implemented current-induced switching experiments in samples with different Pt and Mn thicknesses. We first investigate the impact of $t_{Pt}$ by fixing $t_{Mn}$ to 10 nm and varying $t_{Pt}$ from 0.5 nm to 6 nm. In all the samples, deterministic switching is observed, while the switching polarity remains the same (see Supplementary Note 4). Figure 3a plots $J_c$ as a function of Pt thickness $t_{Pt}$. This trend can be separated into 3 stages: 1) when $t_{Pt} \geq 4$ nm, $J_c$ keeps at a stable plateau; 2) when $1$ nm $\leq t_{Pt} < 4$ nm, $J_c$ decreases with t decreasing $t_{Pt}$; 3) when $t_{Pt} < 1$ nm, $J_c$ starts to slightly increase with decreasing $t_{Pt}$. It is known that the effective SHA of Pt will first increase and then saturate when $t_{Pt}$ increases from 0 nm to 4-5 nm[45]. The change of $J_c$ with $t_{Pt}$ can be understood as follows. At stage 1, STs from Pt dominate the switching and OTs barely participates in the process. At stage 2 and 3, OTs gradually dominate the switching process, since the observed $J_c$-$t_{Pt}$ trend in this region is opposite to the $J_c$-$t_{Pt}$ trend in conventional ST-dominant system (see detailed switching results of $Mn_3Sn/Pt$ in Supplementary Note 2).

The minimum $J_c$ appears at around $t_{Pt} = 1$ nm, revealing that the $L$-$S$ conversion efficiency in $Mn_3Sn/Pt/Mn$ system maximizes at this point. To better quantify the switching efficiency, we measured the effective SHA in $Co/Pt(t_{Pt})/Mn(10$ nm$)$ heterostructure by spin-torque ferromagnetic resonance (ST-FMR) technique (see details in Supplementary Note 7). As shown in Fig. 3b, when $t_{Pt}$ increases from 1 nm to 6 nm, the effective SHA will first increase and then decrease. The highest SHA ($\sim 0.4$) appears when $t_{Pt}$ is around 2-3 nm, which is comparable with other reported values[33,34,41]. We also notice that this optimal SHA point is shifted from the optimal $J_c$ point in $Mn_3Sn/Pt/Mn$ system. A key parameter related to the SOC strength is the spin diffusion length $\lambda_{sf}$. The shorter is $\lambda_{sf}$, the stronger is the SOC. The experimentally determined $\lambda_{sf}$ in $Mn_3Sn$ is $\sim 0.75$ nm[46], which is one order of magnitude shorter than $\lambda_{sf}$ in Co (7-12 nm)[47]. It confirms that the $Mn_3Sn$ has larger SOC than Co, which can explain why we observed deterministic switching in $Mn_3Sn/Cu/CuO_x$ device but we failed to extract spin torque signal in $Co/Cu/CuO_x$ device (Supplementary Note 3). The weaker SOC in Co than that in $Mn_3Sn$ will lead to thicker Pt layer for the

optimal effective SHA, which is the reason of the inconsistency between Co and $Mn_3Sn$ based samples.

We then investigate the impact of $t_{Mn}$ by fixing $t_{Pt}$ to 2 nm and varying $t_{Mn}$ from 3 to 20 nm. As shown in Fig. 3c, a monotonic decreasing $J_c$-$t_{Mn}$ trend can be achieved. When $t_{Mn} = 20$ nm, $J_c$ is reduced to $\sim 1 \times 10^{10}$ A/$m^2$, which is more than one order of magnitude lower than $J_c$ in $Mn_3Sn/Pt$ bilayer[20]. This trend is consistent with the $t_{Mn}$-dependent SHA trend in $Co/Pt/Mn$ system (see Fig. 3d). When $t_{Mn}$ increases from 10 nm to 20 nm, the effective SHA monotonically increases. At $t_{Mn} = 20$ nm, the effective SHA is determined to be around 0.64. We consider this unsaturated effective SHA within a large $t_{Mn}$ range as another important characteristic of OT. According to the drift-diffusion equation, the orbital Hall angle of Mn $\theta_{Mn}$, which is defined as the charge-current-to-orbital-current conversion efficiency, can be described by $\theta_{Mn} = \sigma_{Mn}[1-\text{sech}(t_{Mn}/\lambda_{Mn})]$, where $\sigma_{Mn}$ and $\lambda_{Mn}$ are the orbital Hall conductivity and orbital diffusion length of Mn, respectively. $\lambda_{Mn}$ ($\sim 11$ nm) is theoretically expected to be much longer than the typical spin diffusion length of conventional heavy metal (1-2 nm for Pt)[34]. As a result, $\theta_{Mn}$ should have a much longer saturation length ($> 20$ nm in our work) than the saturation length of effective SHA in Pt (typically 5 nm). Given a fixed L-S conversion efficiency, a larger $\theta_{Mn}$ will certainly lead to a larger effective SHA in the system as well as a lower $J_c$. Note that, we here consider the average current density in Pt/Mn bilayer. In Supplementary Note 8, we show that all the trends and conclusions are still solid regarding the seperated current density distributions in Pt layer and Mn layer. We also show that our OT-driven switching scheme not only reduces $J_c$, but also reduces the switching power consumption by one order of magnitude, which leads to low-power benefit in realistic application (Supplementary Note 9).

To better quantify the actual effective spin Hall angle in our $Mn_3Sn/Pt/Mn$ device, we also implemented harmonics measurement (see measurement setup in inset of Fig. 3e)[48]. When we apply an ac current $I$ along $x$ axis and rotate the external magnetic field in $xz$ plane, the octupole moment $\Delta\varphi_{oct}$ will rotate coherently and result in the change of first harmonic signal $R_\omega$ in xy direction (Fig. 3e). The oscillation of $\Delta\varphi_{oct}$ will also leads to a second harmonic signal $R_{2\omega}$ in the form of $\frac{dR_\omega}{2d\Delta\varphi_{oct}}\big|_{I=0}\Delta\varphi_{oct}(I)$ (Fig. 3f). The current-induced octupole oscillation $\Delta\varphi_{oct}(I)$ can be calculated using the torque balance equation (see calculation details in Supplementary Note 10) The fitting results allow us to obtain the damping-like effective field $H_{DL}$. We can then calculate the effective SHA by $SHA = \frac{2e\mu_0(3M_0)tH_{DL}}{\hbar J}$, where $M_0$, $t$, $\hbar$ and $J_{SOT}$ are the magnetization of a sublattice moment, the $Mn_3Sn$ thickness, the reduced Planck constant and the average current density in the source layer, respectively. In our work, the calculated effective SHA in $Mn_3Sn/Pt(5$ nm$)$ is only $\sim 0.026$. As a comparison, the effective SHA in $Mn_3Sn/Pt(2$ nm$)/Mn(20$ nm$)$ is determined to be $\sim 0.32$, which is more than one order of magnitude higher. This large SHA difference also well corresponds to the $J_c$ difference and demonstrate again that OT-driven $Mn_3Sn$ switching is of high efficiency.

## Neuromorphic computing based on OT-based manipulation of $Mn_3Sn$

Artificial synapses are considered as an ideal hardware to implement neuromorphic computing[49–52]. Recent works suggest the current-induced magnetization switching process in ferro- and ferri-magnetic materials can mimic the long-term depression (LTD) and the long-term potentiation (LTP) functions of synapses, following a general domain nucleation theory[53,54]. However, the linearity of achieved LTD (LTP) processes, which is considered as a fundamental parameter for constructing high-accuracy artificial neural network (ANN), has arrived at a ceiling, because of the limited magnetic domain size. In such case, $Mn_3Sn$ is expected to be a better material platform, since the AFM nature of $Mn_3Sn$ can reduce the magnetic dipole effect and thus allow

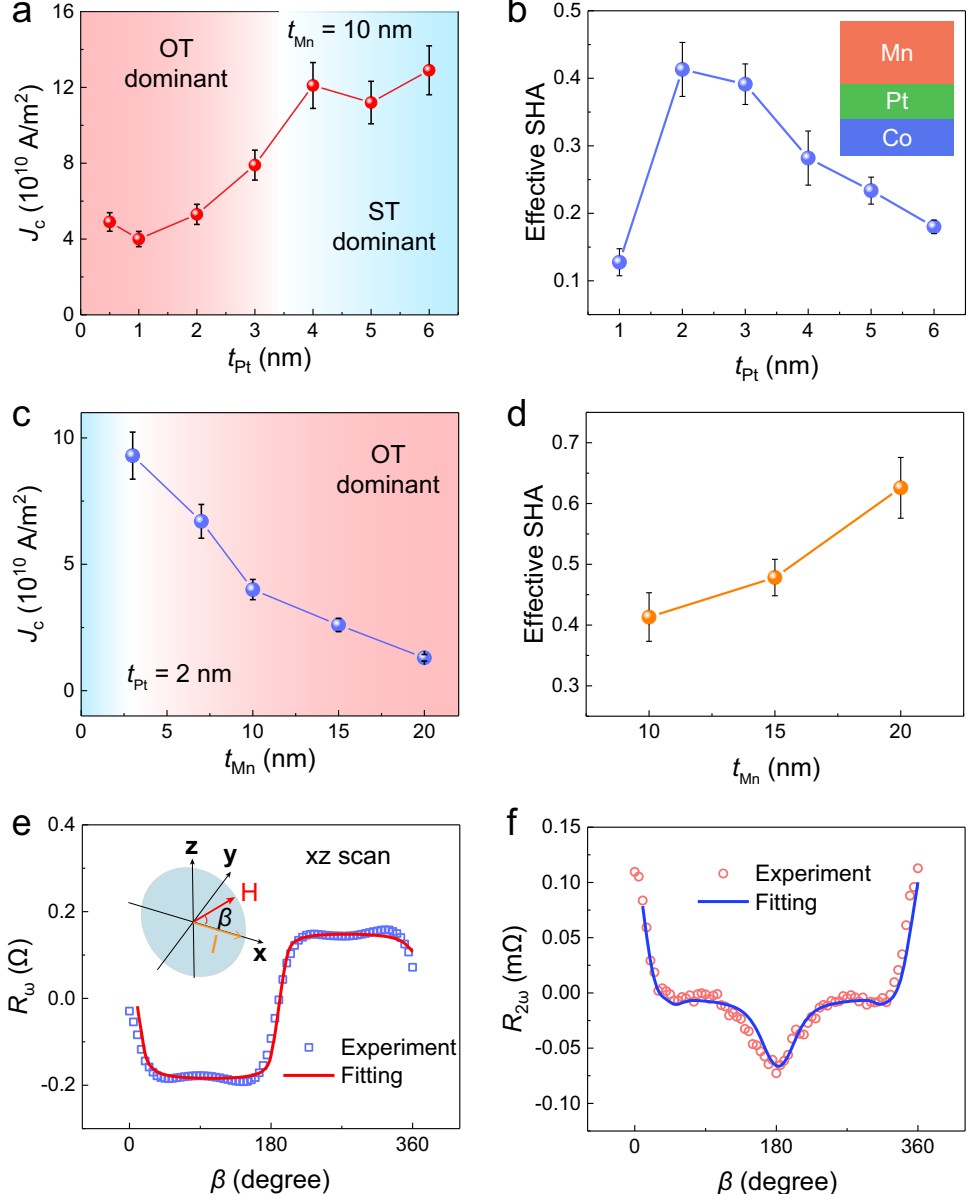

**Fig. 3 | OT source layer dependence of the switching efficiency. a** $J_c$ as a function of Pt thickness $t_{Pt}$ with a fixed Mn thickness ($t_{Mn} = 10$ nm). When $t_{Pt} \leq 3$ nm, the switching dynamics are dominated by OT (red region). When $t_{Pt} > 3$ nm, ST (blue region) dominates the switching dynamics. The error bars in (**a**) and (**c**) are obtained from multiple switching loops for each sample. **b** Effective SHA in Co/Pt/Mn as a function of $t_{Pt}$ while $t_{Mn}$ remains at 10 nm. Inset shows the film stack. The error bars in (**b**) and (**d**) are the standard deviation of SHA at different frequency.

**c** $J_c$ as a function of $t_{Mn}$ with a fixed $t_{Pt}$ (2 nm). A monotonic decreasing trend can be achieved. **d** Effective SHA as a function of $t_{Mn}$ while $t_{Pt}$ remains at 2 nm. **e** First harmonic Hall resistance $R_\omega$ and the fitting curve as a function of $\beta$ in Mn$_3$Sn/Pt(2 nm)/Mn(20 nm) device. The applied magnetic field is 6 T, while the applied current $I$ is 1 mA. *I*nset shows the measurement setup. **f** Second harmonic Hall resistance $R_{2\omega}$ and the fitting curve as a function of $\beta$.

the existence of more tiny magnetic domains in the crossbar area than ferromagnet does. Moreover, the random magnetic domain switching phenomena brought by thermal fluctuation is also expected to be suppressed, given the low OT-driven critical switching current density as well as the good thermal stability of Mn$_3$Sn.

In Fig. 2e, we have already shown that the OT-driven switching process of Mn$_3$Sn exhibits a memristor-like behavior. Here, given the normalized $R_{AHE}$ detected during the switching can be defined as the weight (G) of the artificial synapse, we demonstrate the realization of LTD and LTP functions in a Mn$_3$Sn/Pt(2 nm)/Mn(10 nm) device. As shown in Fig. 4a, to achieve such processes, 120 negative (positive) current pulses of fixed amplitude are applied. The current pulse amplitude is ~$4.5 \times 10^{10}$ A/m$^2$, which corresponds to the very beginning switching point of the device. Interestingly, we found the

achieved LTD (LTP) processes show very high linearity. To better quantify the linearity performance, the nonlinearity (NL) of weight update is defined as $NL = \frac{\max|G_P(n) - G_D(121-n)|}{(G_{max} - G_{min})}$ for $n = 1$ to 120, where $G_P(n)$ and $G_D(n)$ are the normalized R$_{AHE}$ values after the n$^{th}$ potentiation pulse and n$^{th}$ depression pulse. $G_{max}$ and $G_{min}$ represent the maximum $G_P(n)$ after 120 potentiation pulses and minimum $G_D(n)$ at initial state. Here, a very small NL value (~0.166) is determined, indicating that our Mn$_3$Sn-based artificial synapse possesses a similar LTD (LTP) process to an ideal device. As a comparison, if we apply the ST-driven Mn$_3$Sn switching scheme, i.e., Mn$_3$Sn/Pt device, to realize the LTD and LTP process, the obtained NL value (-0.686) is much larger, indicating a worse linearity. See Supplementary Note 11 for detailed discussion.

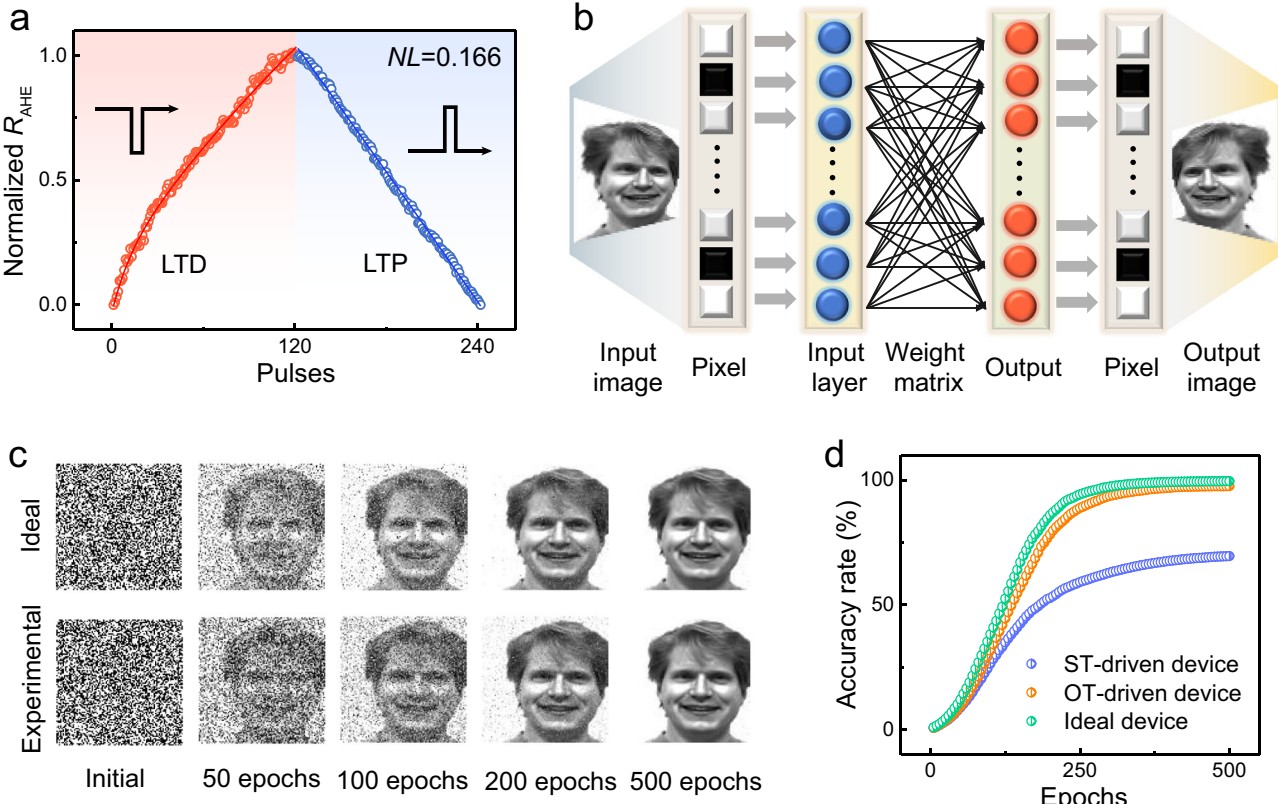

**Fig. 4 | ANN system with Mn₃Sn-based artificial synapse. a** LTD and LTP process with high linearity in Mn$_3$Sn/Pt(2 nm)/Mn(10 nm) device. The calculated *NL* is as low as 0.166, which indicates a good linearity. **b** Schematic of the constructed ANN with 100 × 100 memory cells for image recognition task. **c** Evolution of the images in the learning processes with the experimental and ideal devices. The image is taken from Yale Face Database B[55,56]. **d** Image accuracy rates as a function of learning epochs in constructed ANN using three kinds of devices.

To better evaluate the performance of the proposed synapse, an ANN with a modelled 100 × 100 Mn$_3$Sn-based memory array was then constructed to implement pattern recognition task (see Fig. 4b). The subsequent learning processes followed the synaptic weight change processes shown in Fig. 4a. Each memory cell serves as a synapse to connect pre- and post-neurons, and the synaptic weight of each cell was represented by the gray level of each pixel. The 100 × 100 pixels of initial input image taken from the Yale Face Database B is employed for the pattern recognition task[55,56], and the gray variation of each pixel is real-time stored in each cell with the increasing learning epochs. Figure 4c compares the image evolution with various numbers of learning epochs during the learning process for the experimental and ideal devices. For the quantitative analysis of learning efficiency, the learning accuracy at every 5 epochs, which is defined as the difference between the original image and learned image, can be obtained (see details in Supplementary Note 6). As shown in Fig. 4d, the learning accuracy rate of the OT-driven Mn$_3$Sn device is 97.5%, which is only slightly lower than that of the ideal device (~99.5%). Meanwhile, the lowest learning accuracy is achieved in the ST-driven Mn$_3$Sn device, due to the worst linearity in its LTD and LTP process. The above simulation results strongly suggest the high application potential of our OT-driven Mn$_3$Sn spintronic device in neuromorphic computing. We note that recent studies suggest the presence of memristive behavior in conventional collinear antiferromagnets (AFMs) when subjected to both electrical and optical manipulation[57–59]. On one hand, since the time reversal symmetry (TRS) is conserved, the Néel order in collinear AFMs is usually switched by 90 degrees and remains hard to be detected. In terms of reliable readout, noncolinear AFMs with broken TRS have an advantage over colinear AFMs. On the other hand, we are aware that the ultra-fast switching of noncollinear AFMs at

picosecond timescale hasn't been reported experimentally, despite of theoretical predictions. We expect that the development of effective methods to achieve picosecond manipulation of the Néel order in noncollinear AFMs could further enhance the performance of the constructed ANN.

## Discussion

In summary, we have demonstrated that OTs can serve as an effective electrical method to manipulate the topological magnetization of antiferromagnets. We prove that Mn$_3$Sn can directly convert the orbital current from the OT sources, e.g., metals (Mn) and oxides (CuO$_x$), to spin current. We also show that an inserted Pt layer between Mn$_3$Sn and OT source can enhance the *L-S* conversion efficiency. By adjusting the thickness of Pt and Mn, the critical switching current density can be reduced to as low as -1 × 10$^{10}$ A/m². In addition, we show that the OT-driven switching process in Mn$_3$Sn can mimic the LTD and LTP process with high linearity in an artificial synapse, which can be further utilized to construct ANN system with high image recognition accuracy. Finally, given TMR has been reported in AFM-MTJ, it is possible to incorporate the presented OT-driven Mn$_3$Sn switching scheme in an OT-based AFM MTJ devices. Our finding goes beyond the conventional paradigm of using spin current to manipulate AFM's order, and offers an alternative to integrate topological AFM in future diverse spintronic devices.

## Methods

### Sample growth and device fabrication

Mn$_3$Sn(40)/Pt(0-6)/Mn(0-20)/AlN(5) and Mn$_3$Sn(40)/Cu(10) stacks (thickness in nm) are deposited on thermally oxidized silicon substrates by DC and RF magnetic sputtering (AJA) under a base pressure lower than 3 × 10$^{-8}$ Torr. AlN is an insulating capping layer. Mn$_3$Sn are

deposited by co-sputtering of $Mn_{2.5}Sn$ and Mn targets at room temperature, followed by annealing in situ at 500 °C for 1 hour. After cooling down to room temperature, the following Pt/Mn/AlN or Cu layers are deposited. For the oxidation process, $Mn_3Sn$/Cu films were exposed to dry atmosphere for a set time. Then, AlN is deposited to stop Cu from being further oxidized. The device is fabricated via an Ultraviolet Maskless Lithography machine (TuoTuo Technology). The etching is done by standard ion milling technique.

## Electrical measurement
For transverse and longitudinal signal measurements, an ac current of 317.3 Hz was applied along the x axis, while SR830 and Zurich lock-in amplifiers are used to detect the voltage. For current-induced switching and neuromorphic functions, electrical current pulses (width from 10 to 500 μs) were applied. After each pulse, we wait for 5 s to avoid the Joule heating and use a small ac current to read out the AHE voltage. For ST-FMR measurement, a Rohde & Schwarz SMB 100 A signal generator was used to provide the modulated microwave. The rectifying voltage was collected using a lock-in amplifier.

## Data availability
The authors declare that data supporting the findings of this study are available within the paper and the Supplementary Information file. Further datasets are available from the corresponding author upon request.

## Code availability
The authors declare that code supporting the findings of this study is available within the paper and the Supplementary Information file. Further code is available from the corresponding author upon request.

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

## Acknowledgements
We thank Dr. Zhizhong Zhang and Prof. Yue Zhang for the helpful discussion on the analysis of effective spin Hall angle in $Mn_3Sn$ devices. The research is supported by the Singapore Ministry of Education MOE-T2EP50121-0011, MOE-T2EP50121-0001, MOE Tier 1: 22-4888-A0001, A*STAR RIE2020 Advanced Manufacturing and Engineering (AME) Programmatic Grant- A20G9b0135. The authors acknowledge the use of the Yale Face Database.

## Author contributions
J.S.C., B.K.T, and H.H.L. supervised the project; Z.Y.Z. and J.S.C conceived the idea; Z.Y.Z. deposited the films and fabricated the devices with help from T.Y.Z., H.H.L., L.X.J., Y.D.G. and R.X.; L.Z.R. performed the SQUID experiment; S.S. implemented the XRD measurement; Z.Y.Z. and T.Y.Z. implemented the electrical transport measurement and analyzed the data with help from H.A.Z., Q.H.Z. and J.Q.L.; T.Z. performed the neuromorphic function simulation; T.T.Z., G.L.W. and C.Z. contributed to the data interpretation. Z.Y.Z., T.Y.Z., T.Z. and J.S.C. co-wrote the manuscript. All the authors read and commented on the manuscript.

## Competing interests
The authors declare no competing interests.
