## [Peer Review File · Nature Communications]

Effective electrical manipulation of a topological antiferromagnet by orbital torquesREVIEWER COMMENTS

Reviewer #1 (Remarks to the Author):

Zheng et al. reported an electrical manipulation of the non-collinear antiferromagnet Mn₃Sn in Mn₃Sn/Cu/CuO_x and Mn₃Sn/Pt/Mn heterostructures. The authors claimed that the switching of Mn₃Sn is due to the orbital Hall effect from the CuO_x and the Mn. They also claimed a low switching current and a high effective spin Hall angle as a result of the efficient conversion from charge current to orbital current (eventually to spin current). Comparing this work to existing studies, even if the conclusions drawn by the authors are well supported by the data, I couldn't find a major advancement in terms of fundamental science, nor could I appreciate the improvement of functionality and performance of the devices. I don't think the manuscript meets the standard of Nature Communications. Rather, I think it might be suitable for a more specialized journal such as Communications Physics or Physical Review series. Let me detail my reasoning as follows.

1. This work can be viewed as a combination of two steps, (1) converting the spin current from the orbital current, eventually from the applied charge current via the orbital Hall effect, (2) switching Mn₃Sn by the converted spin current. Regarding (1), there has been experimental studies showing the possible orbital-Hall-effect-induced manipulation of ferromagnetic dynamics (e.g., ref. 40 of the current manuscript, PRB 98, 014401 (2018), Sci. Adv. 4, eaar2250 (2018)) and a direct observation of the orbital Hall effect (Nature 619, 52 (2023), posted on arXiv two years ago). Regarding (2), there has been a number of direct reports. Therefore, neither (1) nor (2) of this manuscript is not new.

2 In some cases, a combination of two known effects can lead to new physics, which can be published in high-impact journals. However, I don't think this situation applies to the current manuscript. From a fundamental point of view, it has been established that a spin current, no matter where it comes from, can be used to switch magnetic moments of the non-collinear antiferromagnet (see the theoretical modeling and experiments of refs. 20 and 21). It is unnecessary to use a complicated structure to prove this fact again.

3. In some other cases, if new functionalities or improved performance is obtained by combining the existing physics, it can also be highly appreciated by the community. However, the switching current density and the effective spin Hall angle reported in the current manuscript cannot compete over existing studies on other materials (see, e.g., Nat.

Mater. 17, 800 and 808 (2018), Nat. Comm. 8, 1364 (2017)). On the other hand, the memristor-like behaviors in the current-induced switching of Mn₃Sn has been shown as well (ref. 20).

4. Apart from the concerns on the novelty and advancement of this work, I am not convinced by the analysis of the origin of the switching. Besides the orbital Hall effect, other possible mechanisms such as the interfacial Rashba spin-orbit coupling at the CuOx/Cu interface and the bulk spin Hall effect of CuOx should be evaluated. Please note that that theoretical predictions (such as ref. 39) can suggest some possible and attracting mechanisms but should not be used as a priori conclusions in experimental studies.

5. Following point 4, the thickness dependence study also needs careful check. The current density in Fig. 3 is defined as the averaged value Pt/Mn. However, Pt and Mn can provide competing mechanisms, that is, spin Hall effect and orbital Hall effect, respectively. Given the different resistivities of Pt and Mn, the actual current distribution and its tendency with thickness cannot be simplified by an averaged density. The authors should check whether the trend in Fig. 3 is affected.

6. I am curious if it is possible to directly quantify the effective spin Hall angle in Mn₃Sn based samples rather than the Mn/Pt/Co. The latter is less helpful for supporting the main argument of this work, the electrical manipulation of Mn₃Sn. On the other hand, with the results obtained in Mn/Pt/Co, the authors had to make some qualitative discussions to explain the inconsistency between Co and Mn₃Sn based samples, which also needs further verification.

Reviewer #2 (Remarks to the Author):

The manuscript deals with the electrical manipulation of the AFM order in Mn₃Sn and reports that the critical current density J_c of the AFM order switching can be reduced to 10^{10} A/m², which is one order of magnitude lower than the common J_c values in SHE-driven framework. Considering that the value of J_c is directly linked with the energy efficiency of device applications, I find this reduction of J_c an important progress. I also expect this method to be very useful for the research field that aims to utilize the AFM for device applications.

Another important message of the manuscript is the claim that the main origin of the J_c reduction is the orbital Hall effect. The experimental data in Figs. 1f and 4 show convincingly that an orbital current is deeply involved with the J_c reduction process. On the other hand, it is not clear to me whether the data in the manuscript can distinguish the orbital Hall effect from the orbital Edelstein effect [33,39]. Not only the orbital Hall effect but also the orbital Edelstein effect can generate an orbital current, which may be converted to a spin current and switch the magnetic order of a given system. So unless the authors can provide experimental evidence to distinguish the two effects, I recommend the authors to be more neutral with regard to the orbital Hall effect vs the orbital Edelstein effect issue.

In addition, I have a few minor comments.

(1) In page 4, it was mentioned that "only when the FM like NiFe exhibits a relatively high SOC, one can observe a sizable effective SHA in FM/Cu/CuOx heterostructure [33,40]". However, Ref. [33] does not examine the FM dependence nor does it examine NiFe, if I am correct. Ref. [40] examined NiFe. But there is a controversy regarding whether NiFe is a good FM material for the orbital-to-spin conversion. For instance, Hayashi et al. [Communications Physics 6, 32 (2023)] reported that the orbital-to-spin conversion is not efficient in NiFe (Fig. 2b). I feel that the authors just want to present an example of a suitable FM for the orbital-to-spin conversion. Then, Ni may be a better example.

(2) At the bottom of page 3, it is mentioned that "As a result, θ_{Mn} should have much longer saturation length (>20 nm in our work) than the effective SHA in Pt (typically 5 nm)." I suspect that the "effective SHA" may be a typo of "spin relaxation length".

To summarize, I find that the manuscript reports a very interesting method to reduce the critical current density of the noncollinear AFM switching. I expect this report will strongly influence the research to utilize the AFM for device applications. It is also very interesting that the reduction is achieved by utilizing an orbital current. Once the authors properly handle a few issues mentioned above, I think this manuscript can be published in Nature Communications.

Response Letter

Reviewer 1

Comments:

Zheng et al. reported an electrical manipulation of the non-collinear antiferromagnet Mn₃Sn in Mn₃Sn/Cu/CuOx and Mn₃Sn/Pt/Mn heterostructures. The authors claimed that the switching of Mn₃Sn is due to the orbital Hall effect from the CuOx and the Mn. They also claimed a low switching current and a high effective spin Hall angle as a result of the efficient conversion from charge current to orbital current (eventually to spin current).

Comparing this work to existing studies, even if the conclusions drawn by the authors are well supported by the data, I couldn't find a major advancement in terms of fundamental science, nor could I appreciate the improvement of functionality and performance of the devices. I don't think the manuscript meets the standard of Nature Communications. Rather, I think it might be suitable for a more specialized journal such as Communications Physics or Physical Review series. Let me detail my reasoning as follows.

Q1. This work can be viewed as a combination of two steps, (1) converting the spin current from the orbital current, eventually from the applied charge current via the orbital Hall effect, (2) switching Mn₃Sn by the converted spin current. Regarding (1), there has been experimental studies showing the possible orbital-Hall-effect-induced manipulation of ferromagnetic dynamics (e.g., ref. 40 of the current manuscript, PRB 98, 014401 (2018), Sci. Adv. 4, eaar2250 (2018)) and a direct observation of the orbital Hall effect (Nature 619, 52 (2023), posted on arXiv two years ago). Regarding (2), there has been a number of direct reports. Therefore, neither (1) nor (2) of this manuscript is not new.

Q2 In some cases, a combination of two known effects can lead to new physics, which can be published in high-impact journals. However, I don't think this situation applies to the current manuscript. From a fundamental point of view, it has been established that a spin current, no matter where it comes from, can be used to switch magnetic moments of the non-collinear antiferromagnet (see the theoretical modeling and experiments of refs. 20 and 21). It is unnecessary to use a complicated structure to prove this fact again.

Authors' Response:

We appreciate the reviewer for the careful review. The instructive comments will definitely help us to improve the quality of this manuscript. We performed additional experiments as well as analysis to address the reviewer's concerns.

Since the main concern of both Q1 and Q2 is about the physical innovation of this manuscript, we response these two questions together. We agree with the reviewer that both orbital-torque-based manipulation of ferromagnetic (FM) order and spin-torque-based manipulation of antiferromagnetic (AFM) order have recently been reported. **However, it still remains unclear whether the orbital torque can be directly used to manipulate AFM order.**

To demonstrate this point, we did current-induced switching experiments in $\text{Mn}_3\text{Sn}/\text{Cu}/\text{CuO}_x$ in the first part of our work. By supplemental experiments addressing Q4 (Figure R4), we show that Cu/CuO_x layer can only generate orbital torques instead of spin torques. To the best of our knowledge, it is **the first experimental demonstration of orbital-torque-based manipulation of AFM order, which should be of broad interests to the spintronic community.** We also argue that orbital torque cannot be naturally considered as an effective manipulation method for all kinds of AFM materials. It should only work for these AFM materials with strong spin-orbital coupling (SOC) which can do the L - S conversion by themselves.

From fundamental point of view, we also want to emphasize that our orbit-torque-based switching experiment not only switches the AFM order, but also **impact the non-trivial topology in Mn_3Sn .** To demonstrate this point, we did supplemental current-induced Anomalous Nerst effect (ANE) signal $V_{2\omega, \text{ANE}}$ switching experiment in $\text{Mn}_3\text{Sn}/\text{Pt}/\text{Mn}$ device, since the large ANE has been proved as a typical transport characteristic of topological property in Mn_3Sn . The switching results are shown in Fig. R1. We can find that $V_{2\omega, \text{ANE}}$ switches simultaneously with the anomalous Hall signal R_ω , suggesting that non-trivial topology in Mn_3Sn can also be manipulated by the orbital torque.

Fig. R1. Current-induced anomalous Hall signal and anomalous Nernst signal switching.

Furthermore, we have also demonstrated the significant improvement in the performance and functionality by using orbital torque to switching non-collinear AFM order, which is shown in the second part of our work, i.e. systematical investigation on the switching by the orbital torque using Pt/Mn bilayer. Please see detailed discussion in the response to Q3.

Modifications in revised manuscript

- 1) We add Fig. R1 and relevant discussion as a new Supplementary Note S5.
- 2) We add description related to Fig. R1 in revised main text (page 5).

3. In some other cases, if new functionalities or improved performance is obtained by combining the existing physics, it can also be highly appreciated by the community. However, the switching current density and the effective spin Hall angle reported in the current manuscript cannot compete over existing studies on other materials (see, e.g., Nat. Mater. 17, 800 and 808 (2018), Nat. Comm. 8, 1364 (2017)). On the other hand, the memrestor-like behaviors in the current-induced switching of Mn₃Sn has been shown as well (ref. 20).

Authors' Response:

We thank the reviewer for the instructive comments. We understand the reviewer's concern. Here, we would like to claim the innovative points from two aspects, i.e., performance and functionality.

1) Performance

The reviewer suggests that our critical switching current density J_c cannot compete over the J_c in topological insulator / CoFeB bilayer. We didn't make such comparison

for the following two reasons. On one side, J_c not only depends on the effective spin Hall angle (SHA), but also depends on the magnetic anisotropic energy to be overcome. Therefore, it is hard for us to compare J_c of 1-nm-thick CoFeB layer with J_c of 40-nm-thick Mn_3Sn layer. From the other side, regarding the uncertainty of the spin transparency at the interface between Mn_3Sn and the applied SOT materials, a novel SOT source layer with a large SHA cannot guarantee a small J_c . Therefore, from our side, we prefer to do a more reasonable comparison using the currently existing or reported experimental data, i.e., a performance comparison between spin-torque-induced switching in Mn_3Sn/Pt and orbital-torque-induced switching in $Mn_3Sn/Pt/Mn$.

In the main text, **we have already reported that our orbital torque strategy can reduce J_c by more than one order of magnitude.** Here, we would like to discuss a more realistic figure of merit for application, which is the switching power consumption P . The calculation is done by using the typical heat dissipation formula $P \propto I_{Pt}^2 R_{Pt} + I_{Mn}^2 R_{Mn}$. The detailed analysis of current distribution in Pt and Mn is shown in the response to Q5. We did the performance comparison in two series of samples. In the first series (Figure R2a), we fixed the thickness of Mn - t_{Mn} to 10 nm and varied the thickness of Pt - t_{Pt} from 0 nm to 6 nm. We normalized P with respect to the switching power of the sample with 6-nm-thick Pt, which can be considered as a totally spin-torque-dominant zone. In the second series (Figure R2b), we fixed t_{Pt} to 2 nm and varied t_{Mn} from 0 nm to 20 nm. We normalized P with respect to the switching power of the sample with 0-nm-thick Mn, which can also be considered as a totally spin-torque-dominant zone. **In both series, when the switching driving force is approaching the orbital-torque-dominant zone, P can be reduced by about one order of magnitude.** We think this systematic comparison can demonstrate the performance superiority of using orbital torque to switch Mn_3Sn .

Fig. R2. **a**, Normalized power as a function of t_{Pt} with a fixed Mn thickness ($t_{Mn} = 10$ nm). **b**, Normalized power as a function of t_{Mn} with a fixed Pt thickness ($t_{Pt} = 2$ nm).

2) Functionality

We thank the reviewer for pointing out that the memristor behavior, or more exactly the multi-state switching behavior shown in Fig. 2e in main text, has been reported in the work [Nature 580, 608-613, 2020]. However, we argue that this memristor behavior is only the precondition for realizing long-term depression (LTD) and the long-term potentiation (LTP) functions, and **it cannot guarantee the achievement of LTD and LTP with high linearity**. To prove the improvement of functionality, we also measured LTD and LTP functions in Mn₃Sn/Pt (5 nm) devices in which spin torque (ST) is the dominant driving force for switching. As shown in Fig. R3a, the achieved nonlinearity (*NL*) of weight update is determined to be 0.686, which is much larger than that in Mn₃Sn/Pt (2 nm)/Mn (10 nm) device and indicates a relatively bad linearity. While a good linearity is of great importance for constructing high-accuracy artificial neural network (ANN), if we use the Mn₃Sn/Pt device to construct an ANN, the corresponding learning accuracy rate is only 69.5% (see Fig. R3b), which is much lower than our Mn₃Sn/Pt (2 nm)/Mn (10 nm) in which orbital torque (OT) is the main driving force for switching.

The relatively bad linearity in Mn₃Sn/Pt device can be explained by the following. According to our power consumption analysis in the performance part, using Mn₃Sn/Pt device to realize LTD and LTP would suffer from a much higher Joule heating effect. Therefore, a certain part of the Mn₃Sn will be switched in a thermally activated mode when the first several pulses are applied, causing an inhomogeneous switching in the pulse sequence, i.e., a worse linearity. As a comparison, using Mn₃Sn/Pt/Mn device to realize LTD and LTP would have lower Joule heating. The switching process in the entire pulse sequence will be more homogeneous, inducing a better linearity. We **thus conclude that orbital-torque-induced Mn₃Sn switching scheme is suitable for the realization of high accuracy ANN**.

Fig. R3. a, LTD and LTP process with bad linearity in Mn₃Sn/Pt device. **b**, Image accuracy rates as a function of learning epochs in constructed ANN using three kinds of devices.

Modifications in revised manuscript

- 1) We add Fig. R2 and relevant discussion as a new Supplementary Note S9.
- 2) We add description related to Fig. R2 in revised main text (page 7).
- 3) We add Fig. R3a and relevant discussion in Supplementary Note S11.
- 4) We replace Fig. 4d in main text by Fig. R3b.
- 5) We add description related to Fig. R3 in revised main text (page 9 and 10).

4. Apart from the concerns on the novelty and advancement of this work, I am not convinced by the analysis of the origin of the switching. Besides the orbital Hall effect, other possible mechanisms such as the interfacial Rashba spin-orbit coupling at the CuO_x/Cu interface and the bulk spin Hall effect of CuO_x should be evaluated. Please note that theoretical predictions (such as ref. 39) can suggest some possible and attracting mechanisms but should not be used as a priori conclusions in experimental studies.

Authors' Response:

We thank the reviewer for pointing this out. We agree that we should distinguish the switching driving force from Cu/CuO_x experimentally. We thus implemented second harmonics measurement in two samples Ni(5 nm)/Cu/CuO_x and Co(5 nm)/Cu/CuO_x. The following is our idea for clarifying the source of the torque: If the switching driving force is indeed the orbital torques, then these two samples should show distinct SOT effective fields, because Ni has stronger SOC than Co. However, if the driving force is the spin torques from interfacial Rashba SOC at the CuO_x/Cu interface or the bulk spin Hall effect of CuO_x as suggested by the reviewer, then these two samples should show similar SOT effective fields.

Fig. R4a shows the measurement setup where we rotate the magnetic field in the *xy* plane with an angle φ to the current direction. Fig. R4b and R4c show the first harmonic signal $R_{\omega}(\varphi)$ and second harmonic signal $R_{2\omega}(\varphi)$ in Ni/Cu/CuO_x sample under $H = 0.5$ T, respectively. The second harmonics signal can be fitted using the following equation suggested in [Nat. Commun., 12, 7111, 2021].

$$R_{2\omega}(\varphi) = \left(R_{\text{AHE}} \frac{B_{\text{DLT}}^y}{B_{\text{eff}}} + R_{\nabla T}^{2\omega} \right) \cos\varphi + 2R_{\text{PHE}} \frac{B_{\text{FLT}}^y + B_{\text{Oe}}}{B_{\text{ext}}} (2\cos^3\varphi - \cos\varphi) - 2R_{\text{PHE}} \frac{B_{\text{DLT}}^z}{B_{\text{ext}}} \cos 2\varphi + R_{\text{AHE}} \frac{B_{\text{FLT}}^z}{B_{\text{eff}}} \quad (1)$$

where B_{eff} , B_{DLT}^y , B_{FLT}^y , B_{DLT}^z and B_{FLT}^z are the effective field, y-polarized damping-like effective field, y-polarized field-like effective field, z-polarized damping-like effective field and z-polarized field-like effective field, respectively. Here, we mainly focus on B_{DLT}^y . Fig. R4d plots $R_{[\cos\varphi]}/R_{\text{AHE}}$ as a function of $1/B_{\text{eff}}$ in Ni/Cu/CuO_x and Co/Cu/CuO_x samples under the same applied current 3 mA. Here, $R_{[\cos\varphi]}$ is the $\cos\varphi$ component obtained by the fitting process, therefore the slope of the fitting curve in Fig. R4d indicates the value of B_{DLT}^y .

Apparently, Ni/Cu/CuO_x sample shows a very steep slope, while the slope of Co/Cu/CuO_x sample is negligible. This comparison reveals that Ni/Cu/CuO_x exhibits a much higher SOT efficiency than Co/Cu/CuO_x, which unambiguously suggest that the main driving torque from Cu/CuO_x bilayer is the orbital torque.

Fig. R4. **a**, Second harmonic measurement setup. **b** and **c**, first harmonic signal $R_{\omega}(\varphi)$ and second harmonic signal $R_{2\omega}(\varphi)$ as a function of φ in Ni/Cu/CuO_x sample under $H = 0.5$ T. **d**, $R_{[\cos\varphi]}/R_{\text{AHE}}$ as a function of $1/B_{\text{eff}}$ in Ni/Cu/CuO_x and Co/Cu/CuO_x samples.

Modifications in revised manuscript

- 1) We add Fig. R4 and relevant discussion as a new Supplementary Note S3.
- 2) We add description related to Fig. R4 in revised main text (page 4).

5. Following point 4, the thickness dependence study also needs careful check. The current density in Fig. 3 is defined as the averaged value Pt/Mn. However, Pt and Mn can provide competing mechanisms, that is, spin Hall effect and orbital Hall effect, respectively. Given the different resistivities of Pt and Mn, the actual current distribution and its tendency with thickness cannot be simplified by an averaged density. The authors should check whether the trend in Fig. 3 is affected.

Authors' Response:

We thank the reviewer for the careful review. We thus recalculated the current density distribution in Pt and Mn layer by the parallel shunting model. We determined the resistivity of Pt to be $\sim 47 \mu\Omega \text{ cm}$ from a 2-nm-thick Pt/AlN device, while the resistivity of Mn was determined to be $\sim 183 \mu\Omega \text{ cm}$ from a 10-nm-thick Mn/AlN device. We note that, when Pt thickness exceeds 5 nm, the resistivity of Pt would reduce a little and lead to a slightly larger calculated current density in Pt in the spin-torque-dominant zone. But it doesn't affect our core discussion at the small Pt thickness range (0-3 nm), i.e., the orbital-torque-dominant zone. The recalculated current density distributions in Pt and Mn corresponding to Fig. 3a and 3c in main text are plotted in Fig. R5a and R5b. We can find that those J_c trends show similar results compared to the trends we obtained by the average density. Therefore, our main conclusion is not affected. And we agree that we should discuss it in our work, so we add it in the supplementary document.

Fig. R5. **a**, Critical switching current distribution in Pt and Mn layers as a function of t_{Pt} with a fixed Mn thickness ($t_{\text{Mn}} = 10 \text{ nm}$). **b**, Critical switching current distribution in Pt and Mn layers as a function of t_{Mn} with a fixed Pt thickness ($t_{\text{Pt}} = 2 \text{ nm}$).

Modifications in revised manuscript

- 1) We add Fig. R5 and relevant discussion as a new Supplementary Note S8.
- 2) We add description related to Fig. R5 in revised main text (page 7).

6. I am curious if it is possible to directly quantify the effective spin Hall angle in Mn₃Sn based samples rather than the Mn/Pt/Co. The latter is less helpful for supporting the main argument of this work, the electrical manipulation of Mn₃Sn. On the other hand, with the results obtained in Mn/Pt/Co, the authors had to make some qualitative discussions to explain the inconsistency between Co and Mn₃Sn based samples, which also needs further verification.

Authors' Response:

We agree with the reviewer that direct quantification of effective spin Hall angle in Mn₃Sn system could better explain our experiment results. Therefore, we implemented harmonics measurement (see measurement setup in inset of Fig. 3e) as proposed by a very recently published work [Nat. Mater. 22, 1106-1113 (2023)].

When we apply an ac current I along x axis and rotate the external magnetic field in xz plane, the octupole moment $\Delta\phi_{oct}$ will rotate coherently and result in the change of first harmonic signal R_ω in xy direction (Fig. R6a). The oscillation of $\Delta\phi_{oct}$ will also leads to a second harmonic signal $R_{2\omega}$ in the form of $\left. \frac{dR_\omega}{2d\Delta\phi_{oct}} \right|_{I=0} \Delta\phi_{oct}(I)$ (Fig. R6b).

The current-induced octupole oscillation $\Delta\phi_{oct}(I)$ can be calculated using the torque balance equation. The calculation process is too complex to be explained here. Please see calculation details in Supplementary Note S10 in revised manuscript. The fitting results allow us to obtain the damping-like effective field H_{DL} . We can then calculate the effective SHA by $SHA = \frac{2e\mu_0(3M_0)tH_{DL}}{\hbar J}$, where M_0 , t , \hbar and J_{SOT} are the magnetization of a sublattice moment, the Mn₃Sn thickness, the reduced Planck constant and the average current density in the source layer, respectively. In our work, the calculated effective SHA in Mn₃Sn/Pt(5 nm) is only ~ 0.026 . As a comparison, the effective SHA in Mn₃Sn/Pt(2 nm)/Mn(20 nm) is determined to be ~ 0.32 , which is more than one order of magnitude higher. This large SHA difference also well corresponds to the J_c difference and demonstrate again that OT-driven Mn₃Sn switching is of high efficiency.

To clarify the inconsistency between Co and Mn₃Sn based samples, we also want to have a little more discussion. A key parameter related to the spin orbital coupling (SOC) strength is the spin diffusion length λ_{sf} . The shorter is λ_{sf} , the stronger is the SOC. The experimentally determined λ_{sf} in Mn₃Sn is ~ 0.75 nm [Phys. Rev. B 99, 184425 (2019)], which is one order of magnitude shorter than λ_{sf} in Co (7-12 nm) [Phys. Rev.

B 98, 174414 (2018); Nanomaterials 11, 2182021 (2021)]. This further confirmed that the Mn₃Sn has larger SOC than Co, which can explain why we observed deterministic switching in Mn₃Sn/Cu/CuO_x device but we failed to extract spin torque signal in Co/Cu/CuO_x device (Supplementary Note S3). The smaller SOC in Co than that in Mn₃Sn will lead to thicker Pt layer for the optimal effective SHA, which is the reason of the inconsistency between Co and Mn₃Sn based samples.

Fig. R6. a, First harmonics Hall resistance R_{ω} and the fitting curve as a function of β . The applied magnetic field is 6 T, while the applied current is 1 mA. Inset shows the measurement setup. **b**, Second harmonics Hall resistance $R_{2\omega}$ and the fitting curve as a function of β .

Modifications in revised manuscript

- 1) We add Fig. R6 as Fig. 3e and 3f in the revised manuscript, while the related description is also added in the main text (page 7).
- 2) The detailed harmonic signal fitting process is added as Supplementary Note S10.
- 3) We add discussions related to the inconsistency between Co and Mn₃Sn based samples in revised main text (page 7).

Again, we would like to sincerely thank the reviewer for the constructive comments. These comments not only allowed us to further understand and enrich our experimental data, but also help us to improve the quality of our manuscript. We hope our response can address your concerns.

Reviewer 2

Comments:

The manuscript deals with the electrical manipulation of the AFM order in Mn₃Sn and reports that the critical current density J_c of the AFM order switching can be reduced to 10^{10} A/m², which is one order of magnitude lower than the common J_c values in SHE-driven framework. Considering that the value of J_c is directly linked with the energy efficiency of device applications, I find this reduction of J_c an important progress. I also expect this method to be very useful for the research field that aims to utilize the AFM for device applications.

Authors' Response:

We appreciate the reviewer for the positive comments of this work, and the constructive suggestions which helped us improve the quality of this manuscript.

Another important message of the manuscript is the claim that the main origin of the J_c reduction is the orbital Hall effect. The experimental data in Figs. 1f and 4 show convincingly that an orbital current is deeply involved with the J_c reduction process. On the other hand, it is not clear to me whether the data in the manuscript can distinguish the orbital Hall effect from the orbital Edelstein effect [33,39]. Not only the orbital Hall effect but also the orbital Edelstein effect can generate an orbital current, which may be converted to a spin current and switch the magnetic order of a given system. So unless the authors can provide experimental evidence to distinguish the two effects, I recommend the authors to be more neutral with regard to the orbital Hall effect vs the orbital Edelstein effect issue.

Authors' Response:

We agree with the reviewer that it is hard to distinguish orbital Hall effect or orbital Rashba Edelstein effect in our sample, particularly in Mn₃Sn/Cu/CuO_x sample. It is indeed much better to be neutral between these two mechanisms. Thus, we use a more general expression *orbital torque* to replace *orbital Hall effect* in the revised manuscript.

Modifications in revised manuscript

- 1) We adapted the expression *orbital torque* to replace OHE in the revised manuscript, which includes the title.

In addition, I have a few minor comments.

(1) In page 4, it was mentioned that "only when the FM like NiFe exhibits a relatively high SOC, one can observe a sizable effective SHA in FM/Cu/CuO_x heterostructure [33,40]". However, Ref. [33] does not examine the FM dependence nor does it examine NiFe, if I am correct. Ref. [40] examined NiFe. But there is a controversy regarding whether NiFe is a good FM material for the orbital-to-spin conversion. For instance, Hayashi et al. [Communications Physics 6, 32 (2023)] reported that the orbital-to-spin conversion is not efficient in NiFe (Fig. 2b). I feel that the authors just want to present an example of a suitable FM for the orbital-to-spin conversion. Then, Ni may be a better example.

Authors' Response:

We totally agree to the reviewer's insightful comments. In fact, in our revised manuscript, we did the second harmonics measurement experiments to demonstrate that Ni/Cu/CuO_x exhibits a much higher SHA than Co/Cu/CuO_x. See details in the response to Q4 for reviewer 1 and Fig. R4. Therefore, we have rephrased our expressions in the revised manuscript.

Modifications in revised manuscript

- 1) We add description related to Fig. R4 in revised main text (page 4).
- 2) We add [Communications Physics 6, 32 (2023)] as a new citation in our main text.

(2) At the bottom of page 3, it is mentioned that "As a result, θ_{Mn} should have much longer saturation length (>20 nm in our work) than the effective SHA in Pt (typically 5 nm)." I suspect that the "effective SHA" may be a typo of "spin relaxation length".

Authors' Response:

We are sorry for the typo. And we did the corresponding modification in the updated manuscript.

Modifications in revised manuscript

- 1) The sentence is rephrased to " θ_{Mn} should have a much longer saturation length (> 20 nm in our work) than the saturation length of effective SHA in Pt (typically 5 nm)."

To summarize, I find that the manuscript reports a very interesting method to reduce the critical current density of the noncollinear AFM switching. I expect this report will strongly influence the research to utilize the AFM for device applications. It is also very interesting that the reduction is achieved by utilizing an orbital current. Once the authors properly handle a few issues mentioned above, I think this manuscript can be published in Nature Communications.

Authors' Response:

We thank the reviewer again for the recommendation of publication in Nature Communications.

REVIEWERS' COMMENTS

Reviewer #1 (Remarks to the Author):

I appreciate the authors' efforts of performing extensive experiments and analysis, which addressed my comments in a satisfactory manner. These additional results highlight the advancement of this work over previous ones and help to clarify some critical concerns from the community on the orbital effects. I am convinced that the revised manuscript can be published in Nature Communications.

The authors should check a possible typo in the schematic of Fig. 3e. According to the AHE data and Fig. S10, the angle β labeled in Fig. 3e should be between I (x axis) and H.

Reviewer #3 (Remarks to the Author):

After reading the manuscript and the response to the first-round referee reports I conclude that the authors have carefully addressed the referees' comments. I also agree with the authors that the work is sufficiently novel and the results inspiring to warrant publication in Nature Communications. I have two minor comments/questions:

1) The authors state: "To better evaluate the performance of the proposed synapse, an ANN with a 100×100 Mn₃Sn-based memory array was then constructed to implement pattern recognition task (see Fig. 4b)."

Did the authors indeed physically construct the memory array as seemingly implied by the above statement? Or, physically, the authors have only discrete devices and the functionality of the 100×100 array was only modelled? I think the authors should be explicit about this point.

2) The authors mention a comparison of the memristive behavior of their non-collinear metallic antiferromagnet with ferromagnetic devices but omit a comparison to previous works on memristive (neuromorphic) devices fabricated from collinear metallic antiferromagnets, including ultra-fast THz and ps-optical multi-level switching. Can the

authors at least briefly discuss the pros and cons of their non-collinear antiferromagnet vs. the earlier studies of collinear antiferromagnets?

Response Letter

Reviewer 1

Comments:

I appreciate the authors' efforts of performing extensive experiments and analysis, which addressed my comments in a satisfactory manner. These additional results highlight the advancement of this work over previous ones and help to clarify some critical concerns from the community on the orbital effects. I am convinced that the revised manuscript can be published in Nature Communications.

The authors should check a possible typo in the schematic of Fig. 3e. According to the AHE data and Fig. S10, the angle beta labeled in Fig. 3e should be between I (x axis) and H.

Authors' Response:

We thank the reviewer for the careful review and for the recommendation of publication in Nature Communications. We also apologize for the typo in Fig. 3e. We have modified the angle beta in a correct way.

Reviewer 3

Comments:

After reading the manuscript and the response to the first-round referee reports I conclude that the authors have carefully addressed the referees' comments. I also agree with the authors that the work is sufficiently novel and the results inspiring to warrant publication in Nature Communications. I have two minor comments/questions:

Authors' Response:

We thank the reviewer for the positive comments and for the valuable questions.

1) The authors state: "To better evaluate the performance of the proposed synapse, an ANN with a 100×100 Mn₃Sn-based memory array was then constructed to implement pattern recognition task (see Fig. 4b)." Did the authors indeed physically construct the memory array as seemingly implied by the above statement? Or, physically, the authors have only discrete devices and the functionality of the 100x100 array was only modelled? I think the authors should be explicit about this point.

Authors' Response:

We thank the reviewer for the care review. Indeed, we only have discrete devices and the 100*100 array was modelled. And we admit that we need to claim it more clearly in the main text. Therefore, in the revised manuscript (Page 7), we add the following description: “To better evaluate the performance of the proposed synapse, an ANN with a modelled 100×100 Mn₃Sn-based memory array was then constructed to implement pattern recognition task (see Fig. 4b).”

2) The authors mention a comparison of the memristive behavior of their non-collinear metallic antiferromagnet with ferromagnetic devices but omit a comparison to previous works on memristive (neuromorphic) devices fabricated from collinear metallic antiferromagnets, including ultra-fast THz and ps-optical multi-level switching. Can the authors at least briefly discuss the pros and cons of their non-collinear antiferromagnet vs. the earlier studies of collinear antiferromagnets?

Authors' Response:

We appreciate the reviewer for valuable question and we agree that adding a brief discussion involving comparison with colinear antiferromagnets is necessary. Therefore, in Page 8, we add a corresponding discussion:

“We note that recent studies suggest the presence of memristive behavior in conventional collinear antiferromagnets (AFMs) when subjected to both electrical and optical manipulation [58-60]. On one hand, since the time reversal symmetry (TRS) is conserved, the Néel order in collinear AFMs is usually switched by 90 degrees and remains hard to be detected. In terms of reliable readout, noncolinear AFMs with broken TRS have an advantage over colinear AFMs. On the other hand, we are aware that the ultra-fast switching of noncolinear AFMs at picosecond timescale hasn't been reported experimentally, despite of theoretical predictions. We expect that the development of effective methods to achieve picosecond manipulation of the Néel order in noncollinear AFMs could further enhance the performance of the constructed ANN.”

We also add several references [58-60] in main text to support this part.